# Comparison of 2D and 3D oxygen-enhanced MRI of the placenta

**Penny L. Hubbard Cristinacce**[1]*, **Minal Patel**[2], **Alexander Oh**[1], **Josephine H. Naish**[1], **Edward D. Johnstone**[2], **Emma Ingram**[2]

**1** Division of Cardiovascular Sciences, Faculty of Biology, Medicine and Health, The University of Manchester, Manchester, United Kingdom, **2** Division of Developmental Biology and Medicine, Faculty of Biology, Medicine and Health, The University of Manchester, Manchester, United Kingdom

* penny.cristinacce@manchester.ac.uk

**Data Availability Statement:** All relevant data and links to the code are within the manuscript and its Supporting Information files. Due to ethical concerns, supporting data cannot be made openly

## Abstract

Oxygen-Enhanced Magnetic Resonance Imaging (OE-MRI) of the human placenta is potentially a sensitive marker of in vivo oxygenation. This methodological study shows that full coverage of the placenta is possible using 3D mapping of the change in longitudinal relaxation rate ($\Delta R_1$), in a group of healthy pregnant subjects breathing elevated levels of oxygen. Twelve pregnant subjects underwent a comparison of 2D and 3D OE-MRI. $\Delta R_1$ was mapped for a single 2D slice (ss-2D), a single matched-slice from the 3D volume (ss-3D) and the full 3D volume (vol-3D). The group-average median $\Delta R_1$ values for ss-3D ($0.023$ s$^{-1}$) and vol-3D ($0.022$ s$^{-1}$) do not differ significantly from ss-2D ($0.020$ s$^{-1}$), when compared using a two-tailed paired t-test (ss-3D ($p = 0.58$) and vol-3D ($p = 0.70$)). However, median baseline $T_1$ ($T_{1b}$) for ss-2D was higher (1603 ms) than $T_{1b}$ for ss-3D (1540 ms, $p = 0.07$) and significantly higher than vol-3D (1515 ms, $p = 0.02$), when compared using a two-tailed paired t-test. In contrast with previous studies, no correlation of median $\Delta R_1$ with gestation age at scan for the normal group ($N = 10$) was observed for ss-2D, likely due to the smaller gestational range. Full volume OE-MRI maps reveal sensitivity to changes in $\Delta R_1$, with some participants showing an enhanced gradient in the intermediate space between the fetal and maternal sides of the placenta in the 3D data. This study shows that it is feasible to acquire whole placental volume OE-MRI data in women with healthy pregnancy.

## Introduction

Oxygen-Enhanced Magnetic Resonance Imaging (OE-MRI) of the human placenta has shown promise in examining placental function [1, 2] and dysfunction [3] by being sensitive to in vivo oxygenation. Abnormalities of placental function can be associated with fetal growth restriction (FGR), which leads to increased neonatal morbidity and mortality [4, 5]. Non-invasive measurements of placental hypoxia, which is associated with peripheral hypovascularity and increased vascular resistance [6], have the potential to identify FGR independent of fetal size.

The spin-lattice relaxation time ($T_1$) of tissue is shortened when a subject breathes an elevated concentration of oxygen due to the paramagnetic effect of the additional dissolved oxygen. This change can be quantitatively assessed by examining the change in $R_1$ (where $R_1 = 1/$

available. Further information about the data and conditions for access are available from the https://qbi-xnat.manchester.ac.uk repository at https://doi.org/10.48420/25266784 for researchers who meet the criteria for access to the data.

**Funding:** This project was funded by Tommy's the baby charity with no specific code and supports numerous people/projects. (EDJ). The funders had no role in study design, data collection and analysis, decision to publish, or preparation of the manuscript.

**Competing interests:** JHN has a shareholding and part time appointment at Bioxydyn Ltd, which provides oxygen-enhanced MRI services. PLHC has worked as a consultant at Bioxydyn Ltd, which provides oxygen-enhanced MRI services. This does not alter our adherence to PLOS ONE policies on sharing data and materials. The other authors have no relevant competing interests.

$T_1$), when a subject is switched from breathing air to 100% oxygen. Previous studies [1–3] have investigated this change by acquiring a single two-dimensional (2D) $T_1$-weighted slice through the placenta. Here we investigate the feasibility of obtaining whole placental oxygenation maps using a three-dimensional (3D) volume acquisition.

The aim of the study was to investigate whether a 3D volume acquisition shows the same quantitative trends in $R_1$ as a single 2D slice, and to assess whether the increased volume covered could offer more sensitivity to placental heterogeneity on a Philips 1.5 T MRI scanner. Full 3D coverage of the placenta may therefore allow a more comprehensive assessment of the placental oxygenation, as well as assessing known changes in gross tissue morphology.

## Methods

Twelve subjects were recruited from St Mary's Hospital, Manchester, following written informed consent (ethical approval REC:14/NW/0195 and 19/NW/0177) between 1st January 2018 to 31st December 2019. A summary of subject demographic and pregnancy outcome is given in Table 1. All subjects had normal uterine artery Doppler flow (pulsatility index <95th centile at 22–24 weeks) and normal umbilical artery Doppler flow (pulsatility index <95th percentile and positive end-diastolic flow) [7]. Pregnancies were subsequently classified as potentially abnormal based on an individualised birth ratio (IBR) <5th percentile (Gestation-related Optimal Weight [8]). Two women did not meet the outcome criteria for assignment to the normal group.

Subjects were scanned using a 1.5T Philips Achieva MRI scanner (Philips Medical Systems Best, NL) whilst supine with a left lateral tilt, to reduce aortocaval compression by the gravid uterus. A cardiac-receiver coil was placed on the abdomen, covering the entire uterus. Non-rebreathing masks (Intersurgical, Wokingham, UK) delivered medical air or 100% oxygen at a flow rate of 15 L/min to the subjects and respiratory triggering was used to minimise motion from maternal breathing.

$T_2$-weighted structural scans were used to determine the position of the placenta and plan the OE-MRI slices. Two inversion recovery OE-MRI protocols were performed: a 3D turbo-spin echo with half Fourier acquisition (3D-HASTE) for full placental coverage and a 2D turbo-spin echo (2D-TSE), as used in previous studies [2, 3]. The single 2D slice positioned

**Table 1. Subject demographic and pregnancy outcomes.**

| Participant No. | Maternal Age Range (years) | BMI | Gestational age at scan (days) | Placental position | Delivery Gestation (days) | Birth weight (g) | IBR (centile) |
|---|---|---|---|---|---|---|---|
| 1 | 31–35 | 24.3 | 196 | Posterior | 266 | 2730 | 10.8 |
| 2 | 16–20 | 29 | 169 | Anterior | 280 | 3289 | 19.7 |
| 3 | 26–30 | 36.9 | 176 | Anterior | 292 | 2900 | 1.3* |
| 4 | 27–30 | 37.9 | 172 | Posterior | 265 | 2900 | 17 |
| 5 | 36–40 | 28.3 | 195 | Posterior | 270 | 3420 | 59.4 |
| 6 | 21–25 | 25 | 169 | Anterior | 282 | 3674 | 41 |
| 7 | 36–40 | 18.1 | 178 | Posterior | 273 | 3682 | 81 |
| 8 | 26–30 | 29.7 | 174 | Posterior | 236 | 2660 | 98 |
| 9 | 26–30 | 26.8 | 170 | Anterior | 275 | 2500 | 1.4* |
| 10 | 31–35 | 17.4 | 176 | Posterior | 286 | 2950 | 12 |
| 11 | 26–30 | 21.3 | 106 | Posterior | 287 | 3106 | 15 |
| 12 | 31–35 | 21.5 | 194 | Anterior | 272 | 2830 | 7 |

*Abnormal based on an individualised birth ratio (IBR) <5th percentile.

perpendicular to the placenta at the level of the cord insertion. The orientation of the 3D-HASTE matched that of the 2D-TSE. For both OE-MRI protocols a $T_1$ map was calculated using a set of inversion recovery images, followed by a dynamic series of $T_1$-weighted images during which the gas supply to the subject is switched from medical air (21%) to 100% oxygen in order to determine $\Delta R_1$.

### 3D OE-MRI protocol

The majority of the data were acquired as follows: $T_1$ mapping data for the 3D-HASTE acquisitions consisted of 5 separate acquisitions with inversion times (TI): 50, 300, 1100, 2000 ms and no inversion pulse ($\approx$TR of 8 s, dependant on triggering), TE of 9 ms, 450 x 450 mm$^2$ FOV, matrix size 176 x 176, voxel size 2.56 x 2.56 mm$^2$, slice thickness 10 mm and 48 slices. Respiratory-triggered $T_1$-weighted images were acquired dynamically at 40 time points with a temporal resolution of ~8 s (triggered off of every other breath), a TI of 1400 ms, with matching TE, FOV, matrix size, voxel size, slice thickness and number of slices. On the 10th dynamic, medical air was switched to 100% oxygen.

On completion of the 3D dynamic acquisition, the supply was switched back to medical air and a 2-minute interval was observed to return the oxygen levels to baseline.

Updated protocol:

In 2 data sets (subjects 11 and 12) the first TI, matrix size and voxel size were changed to match the 2D OE-MRI protocol below (60 ms, 128 x 128, 3.52 x 3.52 mm$^2$).

### 2D OE-MRI protocol

$T_1$ maps for the 2D-TSE acquisitions consisted of 5 inversions times (TI): 60, 300, 1100, 2000 ms and no TI. TR = TI + 8000 ms, TE of 5.4 ms, 450 x 450 mm$^2$ FOV, matrix size 128 x 128, voxel size 3.52 x 3.52 mm$^2$ and slice thickness 10 mm. Respiratory-triggered $T_1$-weighted images were acquired dynamically at 40 time points with a temporal resolution of ~9.4 s and a TI of 1400 ms. On the 10th dynamic, medical air was switched to 100% oxygen.

### Analysis

Data were analysed using in-house Python code. An ROI was defined on the TI = 300 ms image, which incorporated the entire placenta as observed on each slice. Care was taken on the fetal side of the placenta and at each end to exclude areas likely to be affected by motion. A $T_1$ map was created by fitting a 3-parameter inversion recovery model to the data on a voxel-wise basis. The baseline/on air $T_1$ ($T_{1b}$) was used to calculate the change in $R_1$ ($\Delta R_1$) for each dynamic in the $T_1$-weighted series, as described elsewhere(1). The single slice from the 2D OE-MRI (ss-2D) was compared with a single matched slice from the 3D volume (ss-3D) and the entire volume (vol-3D). The slice was matched by using the 3D slice with the closest slice location, according to the DICOM header and checked visually.

Voxel-wise $\Delta R_1$ was calculated by taking the mean of the last 20 dynamics during the oxygen plateau and plotted as maps for both 2D and 3D data. Median and inter-quartile range (IQR) across the ROI of $\Delta R_1$ and $T_{1b}$ were calculated for each subject, to limit the influence of outliers in the voxel-wise maps. The correlation was calculated for median and IQR $\Delta R_1$ and $T_{1b}$ against gestational age at the scan, as in previous reports [1–3]. The significance of the correlation was measured by using the Pearson correlation coefficient and presented as r and *P* values.

Bland–Altman plots were produced to quantitatively assess the agreement between the two methods, with 95% confidence intervals (limits of agreement) calculated as the mean

difference $\pm$ (1.96 × standard deviation of the difference). The ROIs for subjects identified as outliers on these plots were inspected visually on the dynamic data.

Example videos of the 3D $\Delta R_1$ maps can be found in the online supporting information (S4 File).

Two women who did not meet the outcome criteria for assignment to normal group (IBR <5th percentile) were removed from the gestational correlation analysis. All 12 subjects remained in all other comparisons.

## Results

In Fig 1 the subject-average $\Delta R_1$ is plotted against dynamic number for the ss-2D data and the vol-3D data. For both, the signal increases from baseline after the oxygen concentration is increased (10th dynamic) and plateaus by the 20th dynamic image. The ss-2D and vol-3D curves plateaus around the same $\Delta R_1$ with a similar standard deviation on each dynamic $\Delta R_1$.

Table 2 shows the group average median $\Delta R_1$ for ss-2D (0.020 s$^{-1}$), ss-3D (0.023 s$^{-1}$) and vol-3D (0.022 s$^{-1}$) and that ss-3D (p = 0.58) and vol-3D (p = 0.70) do not differ significantly from ss-2D when compared using a two-tailed paired t-test. However, median $T_{1b}$ for ss-2D was higher (1603 ms) than $T_{1b}$ for ss-3D (1540 ms, p = 0.07) and significantly higher than vol-3D (1515 ms, p = 0.02), when compared using a two-tailed paired t-test. Correlation of median $\Delta R_1$ with gestation age at scan for the normal group (N = 10) revealed no trend with gestational age for ss-2D (p = 0.99), ss-3D (p = 0.14) and vol-3D (p = 0.24).

Table 2 also shows the inter-quartile range (IQR) of $\Delta R_1$, a measure of heterogeneity. The ss-2D values are not significantly different from ss-3D (p = 0.94) or vol-3D (p = 0.36) when compared using a two-tailed paired t-test. Correlation of IQR $\Delta R_1$ with gestation age at scan revealed lower values at longer gestation for the 3D data, ss-3D (p = 0.03) and vol-3D (p = 0.005).

A Bland–Altman plot shown in Fig 2 was used to investigate the agreement between $\Delta R_1$ for ss-2D with ss-3D and vol-3D for the individual subjects. The plots show no outliers for ss-

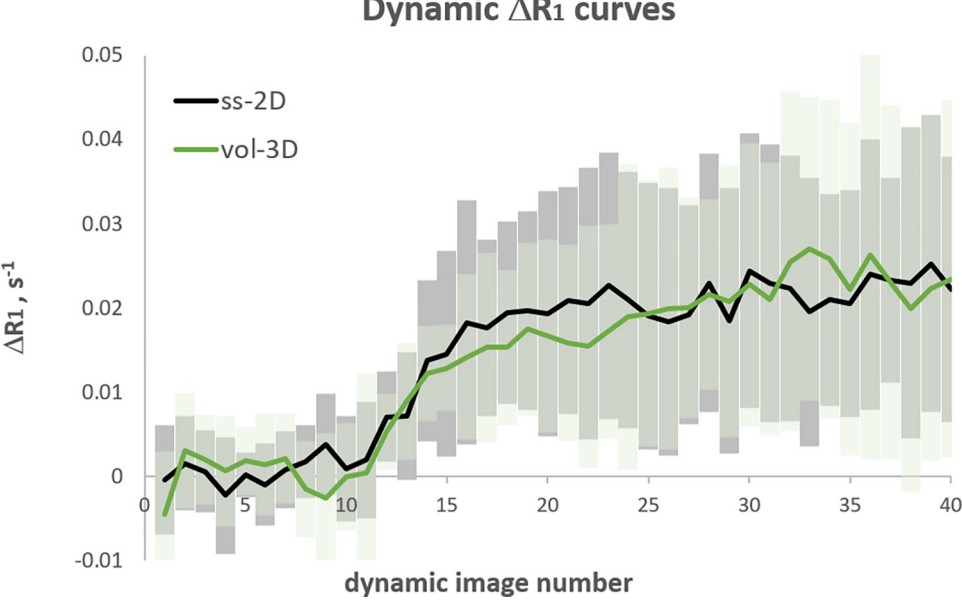

**Fig 1. Mean (line) and standard deviation of the dynamic $\Delta R_1$ curves against dynamic image number for ss-2D (grey) compared with vol-3D (green).**

**Table 2.** Group average median and IQR $\Delta R_1$ and $T_{1b}$ values (12 subjects) and group average median and IQR $\Delta R_1$ and $T_{1b}$ values against gestational age at scan (10 subjects) for ss-2D, ss-3D and vol-3D.

|  | ss-2D | ss-3D | vol-3D |
|---|---|---|---|
| **median $\Delta R_1$, s$^{-1}$** | 0.020 (0.004–0.038) | 0.023 (-0.006–0.047) | 0.022 (-0.003–0.048) |
| **IQR $\Delta R_1$, s$^{-1}$** | 0.024 (0.012–0.048) | 0.025 (0.015–0.045) | 0.027 (0.015–0.038) |
| **median $\Delta R_1$ –gestation** | r 0.00 p 0.99 | r -0.25 p 0.14 | r -0.17 p 0.24 |
| **IQR $\Delta R_1$ –gestation** | r -0.18 p 0.23 | **r -0.45 p 0.03** | **r -0.64 p 0.005** |
| **median $T_{1b}$, ms** | 1603 (1305–1679) | 1540 (1418–1693) | 1515 (1374–1723) |
| **IQR $T_{1b}$, ms** | 177 (86–275) | 196 (122–338) | 217 (152–340) |
| **median $T_{1b}$ –gestation** | r 0.22 p 0.17 | r 0.11 p 0.77 | r 0.00 p 0.99 |
| **IQR $T_{1b}$ –gestation** | r 0.19 p 0.20 | **r -0.39 p 0.05** | **r -0.45 p 0.03** |

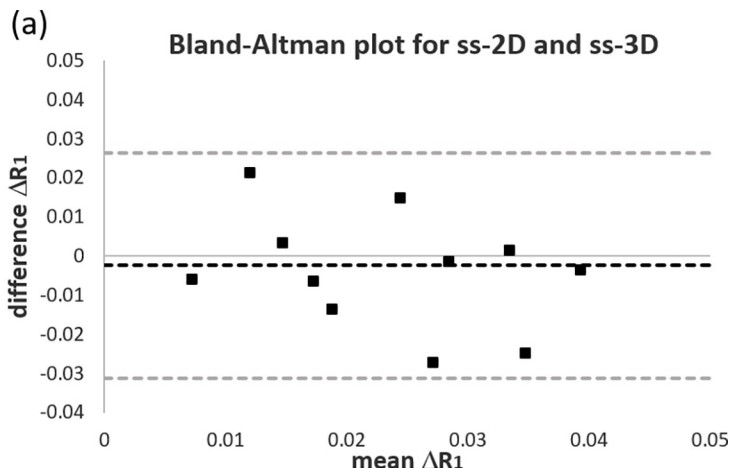

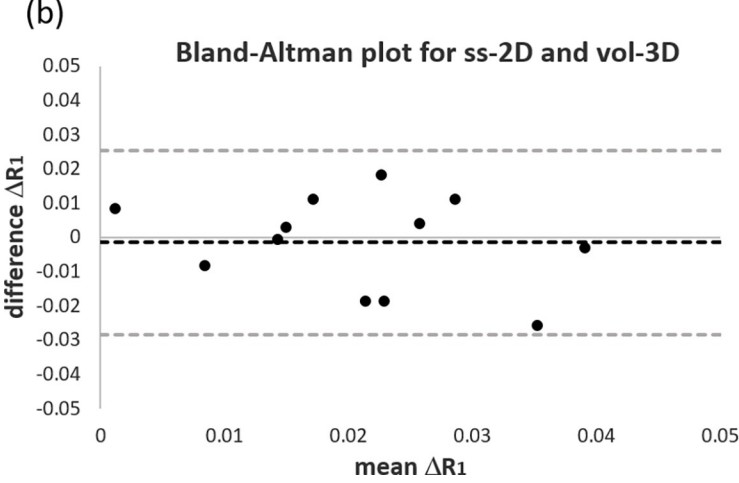

**Fig 2.** Bland–Altman plots of mean $\Delta R_1$ against the difference between 2D and 3D $\Delta R_1$ for (a) ss-2D and ss-3D and (b) ss-2D and vol-3D. The black dotted line represents the mean difference and the grey dotted lines the limits of agreement (95% confidence interval).

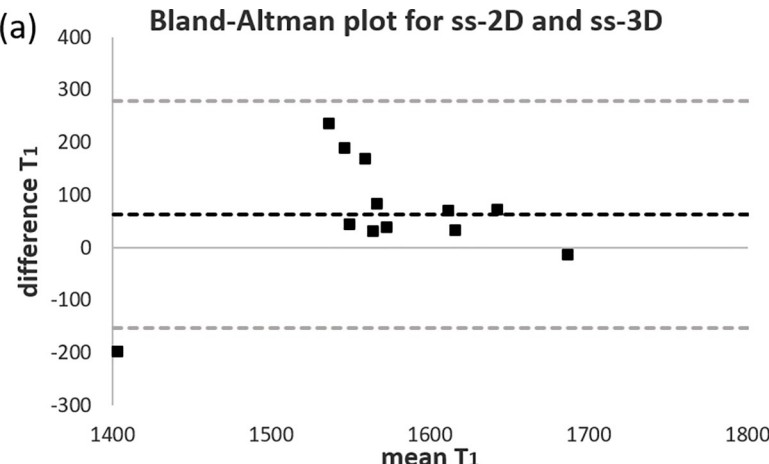

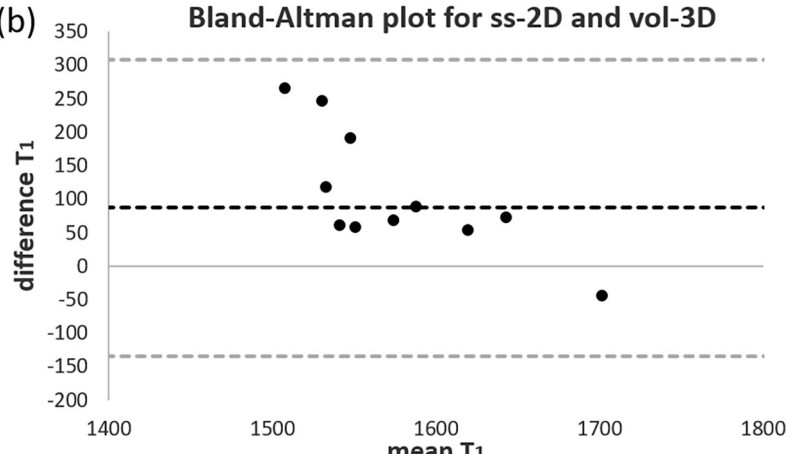

**Fig 3.** Bland–Altman plots of mean $T_{1b}$ against the difference between 2D and 3D $T_{1b}$ for (a) ss-2D and ss-3D and (b) ss-2D and vol-3D. The black dotted line represents the mean difference and the grey dotted lines the limits of agreement (95% confidence interval).

2D and ss-3D or ss-2D and vol-3D. The 2D - 3D difference in $\Delta R_1$ is a smaller magnitude than the mean $\Delta R_1$. The average difference appears to decrease slightly as $\Delta R_1$ increases. Fig 3 show the Bland–Altman plots comparing the $T_{1b}$ values acquired from the 2D and 3D scans. The plots show a single outlier for ss-2D and ss-3D and no outliers for ss-2D and vol-3D. The mean difference shows the trend towards lower $T_{1b}$ values for ss-3D and vol-3D, compared with ss-2D.

Fig 4 shows maps comparing the voxel-wise $\Delta R_1$ for ss-2D and a matching slice from the 3D volume for an example anterior (Fig 4A and 4B) and posterior (Fig 4C and 4D) placenta. The 3D images exhibit similar features to the 2D maps for both of the placenta, however there are amplitude differences for the posterior placenta. Qualitatively, the ss-3D and vol-3D maps are less noisy and more blurred.

## Discussion

This study shows that it is feasible to acquire whole placental volume OE-MRI data on a Philips 1.5 T MRI scanner. All scans were well tolerated by the pregnant subjects, even though

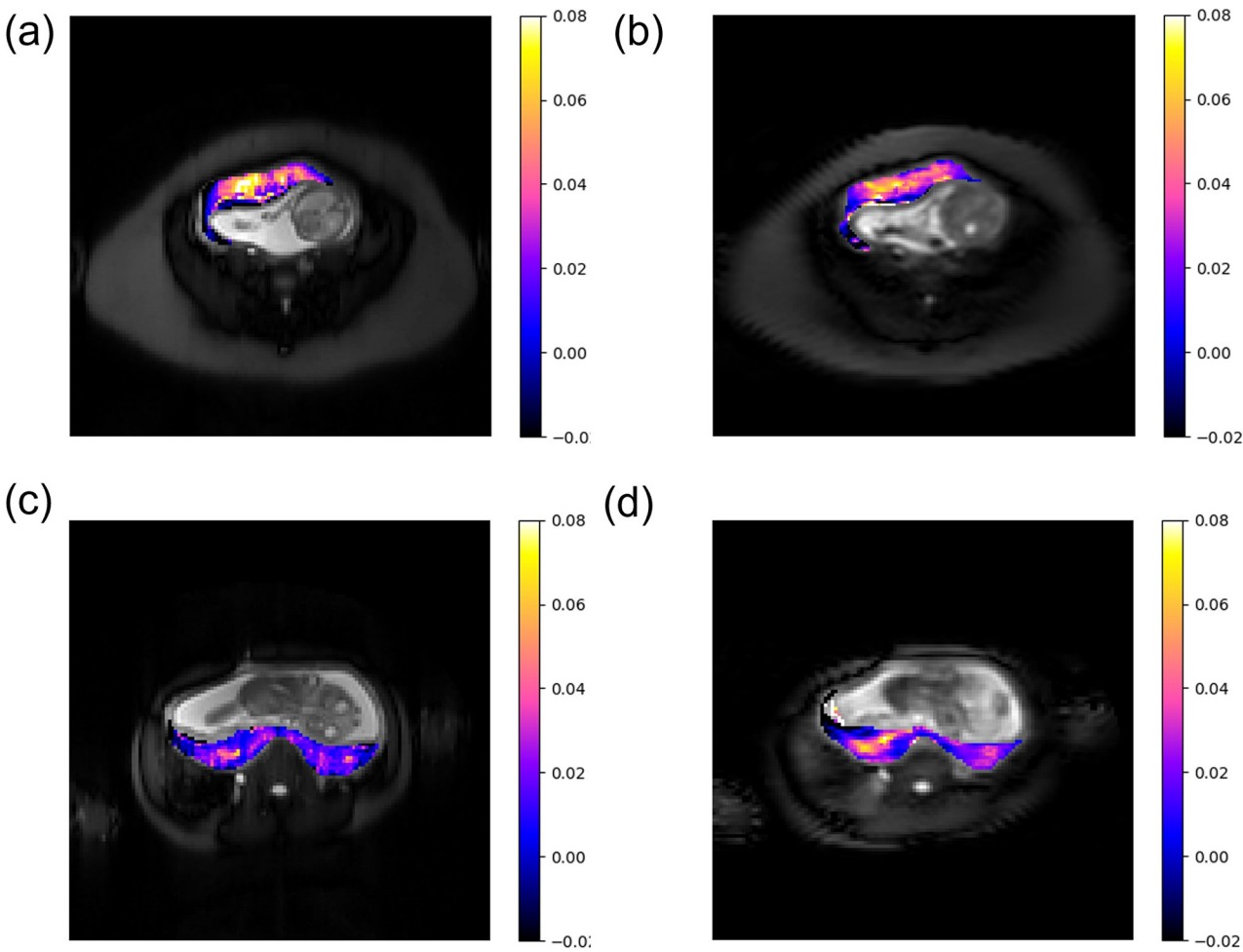

**Fig 4.** *Top*: $\Delta R_1$ $(s^{-1})$ maps of an example anterior placenta for (a) ss-2D and a matching (b) ss-3D; *Bottom*: $\Delta R_1$ $(s^{-1})$ maps of an example posterior placenta for (a) ss-2D and a matching (b) ss-3D.

increasing the coverage to 3D required the use of a cardiac-receiver coil in addition to the non-rebreathing mask.

There were no significant differences between median $\Delta R_1$ for the original single slice 2D protocol (ss-2D) compared with a matched single slice from the 3D data (ss-3D) or the full 3D volume (vol-3D), the individual difference between the 2D and 3D data was on the same order of magnitude as the group average and showed some scatter. All subjects fall within the limit of agreement (95% CI) on the $\Delta R_1$ Bland–Altman plots in Fig 2. Only subject 1 falls outside the limit for the ss-2D v ss-3D comparison of $T_{1b}$ (Fig 3). This subject has $T_{1b}$ in the normal range for ss-3D and vol-3D, but the 2D data shows a low median $T_{1b}$ (ss-2D 1305 ms compared to an average of 1603 ms).

Previous studies have shown a negative correlation of $\Delta R_1$ [1–3] and $T_{1b}$ [9, 10] with gestational age at scan. Here no correlation was significant in this study. However, this study included a relatively small group of subjects (10 compared with between 9 [2] and 41 [10]) and much smaller gestational window (169–196 days compared with up to 145–2671 days [2]). The study cohort was primarily chosen to allow the quality of the data acquired using the 3D protocol to be compared with that of the original 2D protocol, not to test for correlation with gestation. However, a negative trend in IQR $\Delta R_1$ and IQR $T_{1b}$ was apparent for 3D data

(Table 2) corresponding to a reduction in heterogeneity with gestation. The two subjects that were removed from correlation of median $R_1$ with gestational age at scan, due to an IBR < 5%, had relatively low or abnormal $\Delta R_1$. This is in qualitative agreement with the results of [3] (see maps in S4_File–subjects 5 and 9), however the number of subjects is too small to perform a quantitative comparison.

Fig 4 shows that the underlying images are of poorer quality for the 3D data (b and d) than for the 2D data (a and c), with more artefact and significant $T_2$ blurring due to the long echo train and/or undersampling. However, similar features are observed for both the 2D and 3D $\Delta R_1$ maps. $\Delta R_1$ appears to be greater in the intermediate section of the placenta, with a decreasing gradient towards both the fetal and maternal sides. The 3D $\Delta R_1$ video maps in supporting information (S4_File) show that the gradient is observable throughout the extent of the placenta. The higher values of $\Delta R_1$ appear in the intermediate region where the villous tree exchanges oxygen between the maternal and fetal blood [11]. This gradient may be visible as a result of the higher signal-to-noise ratio of the 3D volume acquisition compared with the 2D, but it may be artificially enhanced by partial volume effects at the boundary between the placenta and the surrounding tissue due to image blurring. No statistically significant difference in the heterogeneity measurement IQR between the 2D and the 3D $\Delta R_1$ data was observed, however the correlation of IQR with gestational age shows a statistically significant decrease for both the ss-3D (p = 0.03) and vol-3D (p = 0.005), but not for ss-2D. This may be a result of an overall lower $\Delta R_1$ at later gestation leading to a smaller range or it may reflect a real increase in the homogeneity of oxygenation with increasing gestation to which the 3D data is sensitive.

There are number of limitations of the study. Firstly, the ROI was drawn on the TI = 300ms and applied to the dynamic series of data. The use of the 300ms data for region definition allowed the placenta to be differentiated more clearly, however any gross motion between this scan and the dynamic acquisition used to calculate $\Delta R_1$ could have a detrimental effect on the resulting maps and statistics. Secondly, unlike in previous OE-MRI studies [2, 3] where the ROI has been trimmed to only include voxels where placental tissue was present throughout the whole dynamic, the placental volume for each slice was included without trimming. Care was taken on the fetal side of the placenta to not include approximately one voxel around the amniotic sac to reduce the effects of fetal motion. Any significant motion was accepted as noise on the $\Delta R_1$ curve, with the intention of not unnecessarily removing placental tissue and biasing the analysis by drawing small ROIs. Thirdly, the 3D protocol was updated during the study to match the dimensions of the 2D protocol. This does not appear to have any significant effect on the quality of the data or the resulting maps, but the number of scans with the updated protocol is too small to test for any bias. The group average median $\Delta R1$, with and without participants 11 and 12 (i.e. those scanned using the updated 3D OE protocol), differ by less than 0.002 s$^{-1}$ and both ss-3D (p = 0.58) and vol-3D (p = 0.70) do not differ significantly from ss-2D when compared using a two-tailed paired t-test. This is similar to the data presented for all 12 participants. Finally, the slice matching to obtain the ss-3D data was performed by using the slice location from the DICOM header. This does not take into account gross motion between scans and is only an approximate due to even number of slices in 3D.

The aetiology of FGR is variable with specific phenotypes due to chromosomal, infectious, vascular disease and inflammatory-related causes. It is hoped that whole volume OE-MRI acquisition may allow for in utero placental phenotyping in FGR, thereby improving disease stratification, therapeutic intervention and outcome management. FGR placentas are known to have poor peripheral vascularisation and visible hypo-vascular regions [6]. Other gross lesions evident in maternal vascular malperfusion include placental hypoplasia, infarction and altered villous development [12]. These abnormalities are likely to lead to heterogeneity in the oxygenation across the placental volume that will be more readily observable when imaging

the whole placental volume as opposed to a single central slice. A reliable measure of placental oxygenation across the entire placenta is a welcome addition to the MRI toolbox and could be utilised with other clinical markers and imaging to aid identification of the placental phenotype of FGR and improve fetal outcomes. Here we show that it is possible to obtain a measure of oxygenation via $\Delta R_1$ across the entire placenta on a Philips 1.5 T MRI scanner. The 3D OE-MRI measurement is sensitive to changes in placental oxygenation and allow mapping of the whole placental volume. Further work is necessary to implement this increased coverage on the scanners of other vendors.

## Supporting information

**S1 File. Fig 1 data.** Spreadsheet containing data for the $\Delta R_1$ curves in Fig 1.
(XLSX)

**S2 File. Figs 2 and 3 data.** Spreadsheet containing data for the Bland-Altman plots in Figs 2 and 3.
(XLSX)

**S3 File. Tables 1 and 2 data.** Spreadsheet containing data Tables 1 and 2.
(XLSX)

**S4 File. Fig 4.** Powerpoint file containing videos of ss-2D images and vol-3D videos of voxel-wise ΔR1 maps for all 12 pregnant subjects.
(PPTX)

## Acknowledgments

MRI scans were undertaken within Wolfson Molecular Imaging Centre. We thank radiographers and imaging centre staff for their support.

## Author Contributions

**Conceptualization:** Penny L. Hubbard Cristinacce, Josephine H. Naish, Edward D. Johnstone, Emma Ingram.

**Data curation:** Penny L. Hubbard Cristinacce, Minal Patel, Alexander Oh, Josephine H. Naish, Emma Ingram.

**Formal analysis:** Penny L. Hubbard Cristinacce, Minal Patel, Alexander Oh.

**Funding acquisition:** Edward D. Johnstone, Emma Ingram.

**Investigation:** Minal Patel, Alexander Oh, Josephine H. Naish, Edward D. Johnstone, Emma Ingram.

**Methodology:** Penny L. Hubbard Cristinacce, Josephine H. Naish, Emma Ingram.

**Project administration:** Penny L. Hubbard Cristinacce, Edward D. Johnstone, Emma Ingram.

**Resources:** Josephine H. Naish, Edward D. Johnstone, Emma Ingram.

**Software:** Penny L. Hubbard Cristinacce, Josephine H. Naish.

**Supervision:** Penny L. Hubbard Cristinacce, Josephine H. Naish, Edward D. Johnstone, Emma Ingram.

**Validation:** Penny L. Hubbard Cristinacce, Josephine H. Naish, Edward D. Johnstone, Emma Ingram.

**Visualization:** Penny L. Hubbard Cristinacce, Minal Patel.

**Writing – original draft:** Penny L. Hubbard Cristinacce, Josephine H. Naish, Emma Ingram.

**Writing – review & editing:** Penny L. Hubbard Cristinacce, Minal Patel, Alexander Oh, Josephine H. Naish, Edward D. Johnstone, Emma Ingram.

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
