## [Decision Letter · Decision Letter 0]

27 Dec 2023

PONE-D-23-27377Comparison of 2D and 3D oxygen-enhanced MRI of the placenta.PLOS ONE

Dear Dr. Hubbard Cristinacce,

Thank you for submitting your manuscript to PLOS ONE. After careful consideration, we feel that it has merit but does not fully meet PLOS ONE’s publication criteria as it currently stands. Therefore, we invite you to submit a revised version of the manuscript that addresses the points raised during the review process.

We look forward to receiving your revised manuscript.

Kind regards,

Francesca Crovetto

Academic Editor

PLOS ONE

Journal Requirements:

"This project was funded by Tommy's the baby charity. MRI scans were undertaken within Wolfson Molecular Imaging Centre."

"This project was funded by Tommy's the baby charity with no specific code and supports numerous people/projects.  (EDJ). The funders had no role in study design, data collection and analysis, decision to publish, or preparation of the manuscript."

"JHN has a shareholding and part time appointment at Bioxydyn Ltd, which provides oxygen-enhanced MRI services.

PLHC has worked as a consultant at Bioxydyn Ltd, which provides oxygen-enhanced MRI services.

The other authors have no relevant competing interests."

5. We note that you have indicated that there are restrictions to data sharing for this study. PLOS only allows data to be available upon request if there are legal or ethical restrictions on sharing data publicly. For more information on unacceptable data access restrictions, please see http://journals.plos.org/plosone/s/data-availability#loc-unacceptable-data-access-restrictions. 

Reviewers' comments:

Reviewer's Responses to Questions

**Comments to the Author**

1. Is the manuscript technically sound, and do the data support the conclusions?

Reviewer #1: Yes

2. Has the statistical analysis been performed appropriately and rigorously? 

Reviewer #1: Yes

3. Have the authors made all data underlying the findings in their manuscript fully available?

Reviewer #1: Yes

4. Is the manuscript presented in an intelligible fashion and written in standard English?

Reviewer #1: Yes

5. Review Comments to the Author

Reviewer #1: The manuscript is written well, pointing out its limitations. Comparing the 2D and 3D OE-MRI is an interesting approach to add another possible placental oxygenation methodology. The authors provide complete data on all the 12 samples. In general, I agree with the authors' conclusion in the discussion. However, I would like to make a few comments.

1. Sample size

This is more a question than a comment. The authors have previously published data in OE-MRI and evaluated the feasibility of the 3T scanner (10.1016/j.placenta.2016.01.016), in which they included nine participants. In the current manuscript, they had 12 participants. Was there any sample size calculation or estimation for the present study? Could they obtain more participants from the second 3D protocol? If not, why?

2. Change of the 3D protocol

Sometimes it is inevitable to update a protocol in the middle of a study aiming for better results. However, it is difficult to defense this when there are different sample sizes in each protocol, such as in this study (10 vs 2). I understand that the sample size is limited, and it is challenging to prove it statistically. But I believe it is important to justify more about the effect of changing the protocol (lines 214-216).

3. Comments on the scanner feasibility

I find it a bit strange to read the comments about the feasibility of 3D OE-MRI on a Philips 1.5T MRI scanner in the first sentence of the discussion (line 167) and in the conclusion of the abstract (line 26) since it is not mentioned in the aim or introduction. It is more natural to say it is one of the aims and then comment on it in the discussion or the conclusion. Furthermore, the conclusion in the abstract and the main manuscript seem to have discrepancies.

Minor comment:

Line108: Something like “as in previous reports (1-3)” looks better.

6. PLOS authors have the option to publish the peer review history of their article (what does this mean?). If published, this will include your full peer review and any attached files.

Reviewer #1: No

---

## [Author Response · Author response to Decision Letter 0]

27 Feb 2024

We thank the editor and reviewer for their input and useful suggestions. Please find our response to the points raise below:

Journal requirements

https://journals.plos.org/plosone/s/file?id=wjVg/PLOSOne_formatting_sample_main_body.pdf [journals.plos.org] and 

https://journals.plos.org/plosone/s/file?id=ba62/PLOSOne_formatting_sample_title_authors_affiliations.pdf [journals.plos.org]

The manuscript has now been amended to adhere to the PLOS ONE style guidance. Specifically, the title page has been amended, header sizes have been increased, figure referencing amended, and supporting information files renamed.

2) Note from Emily Chenette, Editor in Chief of PLOS ONE, and Iain Hrynaszkiewicz, Director of Open Research Solutions at PLOS: Did you know that depositing data in a repository is associated with up to a 25% citation advantage (https://doi.org/10.1371/journal.pone.0230416)? [doi.org] If you’ve not already done so, consider depositing your raw data in a repository to ensure your work is read, appreciated and cited by the largest possible audience. You’ll also earn an Accessible Data icon on your published paper if you deposit your data in any participating repository (https://plos.org/open-science/open-data/#accessible-data [plos.org]).

We have now uploaded the image data to our internal XNAT repository: https://qbi-xnat.manchester.ac.uk. To request access researchers should contact the corresponding author.

3) Thank you for stating the following in the Acknowledgments Section of your manuscript: 

"This project was funded by Tommy's the baby charity. MRI scans were undertaken within Wolfson Molecular Imaging Centre."We note that you have provided funding information that is currently declared in your Funding Statement. However, funding information should not appear in the Acknowledgments section or other areas of your manuscript. We will only publish funding information present in the Funding Statement section of the online submission form. Please remove any funding-related text from the manuscript and let us know how you would like to update your Funding Statement. 

The acknowledgements have been amended to remove reference to funding. We are happy with the current Funding Statement.

4) Thank you for stating the following in the Competing Interests section: 

"JHN has a shareholding and part time appointment at Bioxydyn Ltd, which provides oxygen-enhanced MRI services.PLHC has worked as a consultant at Bioxydyn Ltd, which provides oxygen-enhanced MRI services. The other authors have no relevant competing interests."

Please confirm that this does not alter your adherence to all PLOS ONE policies on sharing data and materials, by including the following statement: ""This does not alter our adherence to PLOS ONE policies on sharing data and materials.” (as detailed online in our guide for authors http://journals.plos.org/plosone/s/competing-interests). [journals.plos.org] If there are restrictions on sharing of data and/or materials, please state these. Please note that we cannot proceed with consideration of your article until this information has been declared. Please include your updated Competing Interests statement in your cover letter; we will change the online submission form on your behalf.

Please update the Competing Interests section to read: "JHN has a shareholding and part time appointment at Bioxydyn Ltd, which provides oxygen-enhanced MRI services. PLHC has worked as a consultant at Bioxydyn Ltd, which provides oxygen-enhanced MRI services. This does not alter our adherence to PLOS ONE policies on sharing data and materials. The other authors have no relevant competing interests."

5) We note that you have indicated that there are restrictions to data sharing for this study. PLOS only allows data to be available upon request if there are legal or ethical restrictions on sharing data publicly. For more information on unacceptable data access restrictions, please see http://journals.plos.org/plosone/s/data-availability#loc-unacceptable-data-access-restrictions. [journals.plos.org] Before we proceed with your manuscript, please address the following prompts:

https://journals.plos.org/plosone/s/recommended-repositories [journals.plos.org]. You also have the option of uploading the data as Supporting Information files, but we would recommend depositing data directly to a data repository if possible. We will update your Data Availability statement on your behalf to reflect the information you provide.

Data contain potentially identifying or sensitive patient information of pregnant participants. The manuscript details the location of the scanning, the approximate date and the gestation of the pregnancy, and as such we are concerned about the ability of the participants to be identified via malicious data linking if imaging data are freely available. The Participant Information Sheet, agreed by the Ethics Committee, states that “Your information will only be used by organisations and researchers to conduct research in accordance with the UK Policy Framework for Health and Social Care Research (https://www.hra.nhs.uk/planning-and-improving-research/policies-standards-legislation/uk-policy-framework-health-social-care-research/).” As such, we feel it appropriate to check who is using the raw imaging data.

These images have now been uploaded to an internal XNAT repository. Requests for access should be sent to the corresponding author. All processed data which was used to create the figures in this manuscript is available as Supporting Information.

6) Please include captions for your Supporting Information files at the end of your manuscript, and update any in-text citations to match accordingly. Please see our Supporting Information guidelines for more information: http://journals.plos.org/plosone/s/supporting-information. [journals.plos.org]

The supporting information file captions for now been included and file names amended accordingly.

The references have been reviewed and doi added throughout.

Reviewer requirements

1) Sample size. This is more a question than a comment. The authors have previously published data in OE-MRI and evaluated the feasibility of the 3T scanner (10.1016/j.placenta.2016.01.016), in which they included nine participants. In the current manuscript, they had 12 participants. Was there any sample size calculation or estimation for the present study? Could they obtain more participants from the second 3D protocol? If not, why?

There was no sample size calculation performed for this study and the number of participants was limited by the timescale of the MSc students who were involved in the recruitment process. However, as the reviewer correctly states, previous studies used fewer participants, so we believe this number to be sufficient.

2) Change in the 3D protocol. Sometimes it is inevitable to update a protocol in the middle of a study aiming for better results. However, it is difficult to defense this when there are different sample sizes in each protocol, such as in this study (10 vs 2). I understand that the sample size is limited, and it is challenging to prove it statistically. But I believe it is important to justify more about the effect of changing the protocol (lines 214-216).

We agree that it isn’t ideal that the imaging protocol was changed. On inspection of the results with participants 11 and 12 removed (those with the updated 3D protocol), we observed a group average median �R1 for ss-2D (0.019 s-1), ss-3D (0.021 s-1) and vol-3D (0.021 s-1) and that ss-3D (p = 0.72) and vol-3D (p = 0.76) do not differ significantly from ss-2D when compared using a two-tailed paired t-test. This compares with the following (as stated in the manuscript):

“Table 2 shows the group average median �R1 for ss-2D (0.020 s-1), ss-3D (0.023 s-1) and vol-3D (0.022 s-1) and that ss-3D (p = 0.58) and vol-3D (p = 0.70) do not differ significantly from ss-2D when compared using a two-tailed paired t-test.” 

We have added the following statement to the discussion of the manuscript:

“The group average median �R1, with and without participants 11 and 12 (i.e. those scanned using the updated 3D OE protocol), differ by less than 0.002 s-1 and both ss-3D (p = 0.58) and vol-3D (p = 0.70) do not differ significantly from ss-2D when compared using a two-tailed paired t-test. This is similar to the data presented for all 12 participants.” 

3) Comments on the scanner feasibility. I find it a bit strange to read the comments about the feasibility of 3D OE-MRI on a Philips 1.5T MRI scanner in the first sentence of the discussion (line 167) and in the conclusion of the abstract (line 26) since it is not mentioned in the aim or introduction. It is more natural to say it is one of the aims and then comment on it in the discussion or the conclusion. Furthermore, the conclusion in the abstract and the main manuscript seem to have discrepancies.

We have looked to make the abstract, aims and discussion more coherent. The text “… placenta on a Philips 1.5 T MRI scanner. Further work is necessary to implement this increased coverage on the scanners of other vendors.” has been removed from the abstract and replaced with “…in women with healthy pregnancy”. The text has been added to line 48 of the introduction, as an aim, and to line 240 of the discussion, as future work. See highlight text in updated manuscript.

4) Minor comment:

Line108: Something like “as in previous reports (1-3)” looks better.

Minor comment addressed.

We hope the amendments described are satisfactory and reach the expected standard for the journal.

---

## [Editor Report · Decision Letter 1]

9 Apr 2024

Comparison of 2D and 3D oxygen-enhanced MRI of the placenta.

PONE-D-23-27377R1

Dear Dr. Penny,

We’re pleased to inform you that your manuscript has been judged scientifically suitable for publication and will be formally accepted for publication once it meets all outstanding technical requirements.

Kind regards,

Francesca Crovetto

Academic Editor

PLOS ONE

---

## [Editor Report · Acceptance letter]

1 May 2024

PONE-D-23-27377R1 

PLOS ONE

Dear Dr. Hubbard Cristinacce, 

I'm pleased to inform you that your manuscript has been deemed suitable for publication in PLOS ONE. Congratulations! Your manuscript is now being handed over to our production team.

Kind regards, 

on behalf of

Dr. Francesca Crovetto 

Academic Editor

PLOS ONE